# Dairy farmer practices and attitudes relating to pasture-based and indoor production systems in Scotland

**Orla K. Shortall**[1]*, **Altea Lorenzo-Arribas**[2]

**1** Social, economic and geographical sciences, James Hutton Institute, Aberdeen, United Kingdom,
**2** Biomathematics and Statistics Scotland, Aberdeen, United Kingdom

* orla.shortall@hutton.ac.uk

## Abstract

This study assesses the practices and views of Scottish dairy farmers relating to pasture-based and indoor systems. There are the debates about the environmental, economic and animal welfare implications of these systems. Indoor dairy farming is a contentious practice among the public. While this controversy is sometimes represented as a lack of public understanding, there is a need for more research on farmers' views to facilitate discussion in the industry. A survey was posted to 909 dairy farmers in Scotland with questions about their grazing practices and attitudes to grazing and indoor systems. 254 surveys were completed, online and in paper form. There was a 26% response rate to the paper version of the survey. The results showed that 19% of respondents housed some or all the cows all year-round. 68% agreed or strongly agreed that cows should graze for part of the year and 51% agreed or strongly agreed that welfare was better if cows grazed for part of the year. These views coexisted with the view that management was more important than the type of system for determining profitability or welfare outcomes (83% and 82% strongly agree or agree respectively). Respondents whose system involved grazing and respondents who had spent longer in farming were moderately more likely to agree that cows should have access to pasture, and slightly less likely to agree that management was more important than system for determining welfare outcomes. The results indicate that the picture is more complicated than the public rejecting indoor dairy farming and those in the industry accepting it. The results showed that a majority preference for cows to graze co-existed with the view that management was more important than system. In terms of industry and policy recommendations, the research suggests that measures should be taken to safeguard farmers' ability to graze through for instance research and advisory support on grazing; ensuring different systems are not penalised in the development of dairy sector environmental measures and recommendations; and potentially supply chains that financially rewards farmers for grazing.

**Data Availability Statement:** All relevant data files are available in the UK Data Service public repository: https://reshare.ukdataservice.ac.uk/855220/.

**Funding:** OS was supported by a British Academy Postdoctoral Fellowship (https://www.thebritishacademy.ac.uk/). ALA was supported by Scottish Government Rural and Environment Science and Analytical Services Division, as part of the Centre of Expertise on Animal Disease Outbreaks (EPIC). Fieldwork costs were supported by the British Academy and Scottish Government Rural and Environment Science and Analytical Services Division, as part of the Centre of Expertise on Animal Disease Outbreaks (EPIC). The funders had no role in study design, data collection and analysis, decision to publish, or preparation of the manuscript.

**Competing interests:** The authors have declared that no competing interests exist.

# Introduction

Dairy farming in industrialised countries has undergone a process of consolidation for decades with fewer, larger and more productive herds [1]. Grazing and forage feeds have decreased in importance with more non-forage feedstuffs such as concentrate and cereals used to increase yields [2]. The number of dairy farms where cattle graze and the amount of time cows spend grazing has declined in recent decades in countries in Europe, including the UK [3]. There has been public opposition to the consolidation of the dairy sector in the UK and the year-round housing of cows [4]. Research has been carried out with the public on views about indoor dairy farming in the UK [5–7], and UK industry stakeholders [8] but none with dairy farmers themselves. It is important to understand dairy farmer attitudes and practices in order to incorporate their views into debates about controversial areas within agriculture [9].

The terms 'pasture-based' or 'grass-based' system are used in this paper to refer to systems where cows graze for part of the year. These systems may involve year-round grazing but usually involve a period of housing cows in winter in temperate countries. An indoor system means that the cows are housed all year-round and do not graze. Scotland is an interesting case study to explore farmer attitudes to pasture-based and indoor systems because the Scottish dairy sector is characterised by a diversity of systems [10]. And Scotland's dairy industry is moving towards fewer, larger herds [11]. There is a correlation between larger herds and year-round housing [12] and smaller herds can save costs by operating a grazing system [13]. Thus, questions around grazing and housing have a bearing on the future structure of the Scottish dairy industry. Findings from this research can also provide insights for other countries in the UK and countries with diverse grazing and housing practices.

Around a third of farmers in the UK use the traditional system of summer grazing and winter housing, whereas the rest are housing or feeding cows indoors for more of the year [12]. Estimates of dairy farmers housing all or some of the cows in the UK range from 16% [12] to 23% [14]. Grass is becoming less important in the production of milk in the UK: a survey with 2000 farms in the UK showed that milk yield from grazed grass had decreased between 2009 and 2019 [13]. Year-round housing allows farms to expand beyond the limits of their grazing platform, to increase yields through feeding more energy dense feed indoors and/or to have greater oversight of the health and activities of the cows [15].

There has been debate about the animal health and welfare implications of housing cows all year-round. Evidence has suggested that indoor dairy farming can result in worse health outcomes for cows including lameness and mastitis [16, 17] and cows show some preference for spending time outside when given the choice [16, 18]. A study suggested grazing was beneficial for cows' emotional wellbeing [19], and another that cows within the same herds had better welfare when they were grazing in summer compared to being housed in winter [20]. It is claimed that year-round housing can also be detrimental for welfare because it does not allow cows to express natural grazing behaviour and can stifle social behaviours [21]. These claims are disputed within the dairy industry, with qualitative research showing key industry stakeholders claimed welfare outcomes were not predicated on the type of system, but management and facilities were more important [8].

The environmental aspects of pasture-based and indoor systems are also debated. It is claimed that systems involving a high proportion of grass in the diet have lower greenhouse gas emissions per litre of milk because of carbon sequestered in grassland [22, 23]. There have been criticisms from the media and charities that purchased feed for livestock is an inefficient use of resources and leads to environmental degradation in its location of production [24].

In defence of the environmental credentials of high-feed-input systems, others point out that intensification through more bought in feed could reduce emissions per litre of milk: as

production per cow increases, greenhouse gases per litre of milk may decrease because fewer cows are needed to produce the same amount of milk, meaning a relative reduction in methane emissions [25].

A report in 2021 by the Scottish Dairy Sector Climate Change Group detailed actions to lower greenhouse gas emissions across different types of systems in the Scottish dairy sector [26]. In relation to grazing the report suggested better grass management in terms of use of cover crops, legumes, high sugar grasses and avoiding soil compaction, rather than a system change through increasing the proportion of grazed grass in the diet.

In relation to the economic aspects of systems that involve grazing or no grazing, the dominant view among key stakeholders in the UK is that economic outcomes are not determined by the type of system, but management plays a more important role [8]. A government-industry report about the future of the dairy industry from 2014 states: "Our evidence shows that system and herd size are not predictors of profitability. Any system of any size, run well, can be profitable and sustainable." (20 p.14). Qualitative research with key stakeholders found that there was a marginal view in the UK favouring the economic benefits of systems that aim to maximise milk production from grass, because grass is the lowest cost feedstuff [8].

Research has shown that the majority of the public in the UK are opposed to indoor dairy farming. Ellis et al. (2009) found that 95% of public respondents stated that they did not think it was acceptable to keep cows inside all year-round; a YouGov poll commissioned by the non-governmental organisation World Animal Protection found that 86% of respondents agreed that cows should graze [7]; and a YouGov poll carried out by the Free Range Dairy Network found 74% of respondents were prepared to pay more in coffee shops for milk from cows that had spent time outdoors [27]. A study with UK citizens found that they ranked access to grazing, cow comfort; and health and welfare as their top priorities [5]. Within the dairy industry in the UK it is stated that public opposition may stem from a lack of understanding and familiarity with indoor dairy farming [28]. A qualitative study of the views of key stakeholders found that the dominant discourse in the mainstream UK dairy industry was that system differences did not matter for determining economic, animal welfare and environmental outcomes, but rather on-farm management was seen as key [8].

There has been research with farmers in Germany, Denmark and Canada about their attitudes towards grazing, both countries where the majority of farms house cows all year-round [29–32]. These studies found that farmers whose system involved grazing had more positive attitudes towards grazing, emphasising lower feed costs, lower labour input, improvements in cow health and fertility [30], public image [31] and suitability to the local climate and existing infrastructure, better animal welfare, easier management outdoors and a price premium based on pasture access [32]. Respondents whose system did not involve grazing had more negative attitudes towards grazing, emphasising lower yields, lower profits and difficulties in grazing a larger herd [30–32] and adverse climate conditions [32]. Another study in Germany found that older farmers had more negative attitudes towards grazing than younger farmers [29]. Research has found differences between the views of farmers and citizens on the issue of grazing or year-round housing. A qualitative study in Brazil compared the views of farmers, agricultural advisors and farmers on their ideas of an ideal dairy farm [33]. Use of pasture was part of the ideal dairy farm for the three groups, but for different reasons: for economic reasons for the farmers and advisors and animal welfare reasons for the citizens. Similarly, a survey study in Belgium found that citizens rated outdoor access and ability to express natural behaviour as more important than did farmers [34]. A study in Canada with citizens, producers, vets, animal advocates and students found a high level of support among dairy producer respondents for access to pasture, though producers only made up 8.7% of their sample [35].

Initiatives have been started in the UK to market milk based on grazing credentials, including The Pasture Fed Livestock Association and the Free Range Dairy Network [36, 37]. Several supermarkets sell liquid milk only from grass-fed cows such as Marks and Spencer, Waitrose and the Co-op, as well as a 'Pasture Promise' label in Asda. Qualitative research with key stakeholders in the UK dairy sector showed some reluctance to market milk based on grazing credentials because it was seen to divide the industry, promote grazing systems over indoor systems and cause confusion among consumers [8].

The future role of grazing and indoor systems in the marketing of Scottish milk is unclear. The Scottish Dairy Review: 'Ambition 2025' and the Dairy Action Plan sets targets for increasing milk production by 50% over 10–12 years, ensuring the resilience and profitability of the Scottish dairy industry, and establishing a distinct market identity for Scottish dairy produce [38]. A Scottish dairy brand was launched in 2015 to differentiate Scottish produce on domestic and export markets [39]. The messaging around the brand did not focus on the provenance of dairy production in terms of grazing or not grazing, rather emphasising added value processed produce, and aiming to capitalise on Scotland's reputation for quality food production (10).

The purpose of this study is to establish up to date evidence about the types of systems farmers are operating in Scotland; to explore their reasons for choosing systems and their attitudes towards indoor and pasture-based systems, in order to inform government and industry policy. Within Scotland, given the diversity of production systems and the dominant discourse that systems differences are not the most important factor for determining outcomes [8] it was hypothesized that the majority of respondents would not endorse the view that cows should graze for part of the year. The aim of the project in addressing these research questions was achieved.

## Methods

### Data collection

The survey included questions about farmer demographic details, including gender, time in farming and ownership structure. Questions about production system included whether the farm was organic, number of cows, area of land, milk yield, calving practices, grazing and housing practices, number of labour units, age of buildings, whether the farm expanded production since 2015 and whether there were plans to expand in the near future. Farmers who had expanded and/or planned to expand were asked about the means of expansion. Those who operated a year-round housing system were asked to rank why they had chosen to this system. The survey is included as supporting information S1 Text.

Attitudinal questions covered attitudes to welfare, environmental and economic aspects of pasture-based and indoor systems using a Likert scale with strongly agree, agree, neither agree nor disagree, disagree or strongly agree options. Respondents were also asked to rate their satisfaction with profitability and work life balancing using a Likert scale of very satisfied, satisfied, neither satisfied nor dissatisfied, dissatisfied and very dissatisfied. There was a question ranking challenges facing the dairy sector and an open question for respondents to leave any additional comments. Ethical approval for the study was gained from the James Hutton Institute research ethics committee. The survey was pilot tested with stakeholders in the Scottish dairy sector.

Contact details for Scottish farmers were obtained from the Scottish government. A paper copy of the survey was posted to 909 dairy farms in Scotland in September 2018 with a cover letter explaining the purpose of the survey and a prepaid return envelope. There was also an online version of the survey which was disseminated on social media and the link was included

in the letter posted to farmers. A donation of £2 was made to a charity supporting farmer well-being for every survey completed, as a gesture of goodwill for farmers who completed the survey. The paper copies of the survey were entered into the online survey platform by a data entry company.

### Data analysis

Data was cleaned to by assessing continuous variables (i.e., number of cows, amount of land, milk yield and number of labour units) graphically for the presence of outliers by means of boxplots that highlighted the presence of high numbers considered to be anomalies (not consistent with the overall population) that were obvious consequences of input errors and therefore were removed. Descriptive statistical analysis of farmer demographic and attitudinal data was carried out using Stata 15.0 [40]. In order to assess whether there were underlying groups of farmers with similar demographics and attitudinal responses, cluster analysis was run using R 4.0.3 [41]. Gower distance was used to account for the different types of variables (categorical, ordinal and numerical) in the data set. Subsequently, to determine the specific factors affecting answers to attitudinal questions about pasture-based and indoor systems, ordinal regression models were fitted using Stata 15.0. Calving pattern was included as a variable in the models because it is an important aspect of the farm system with different calving patterns associated with different levels of cost, farmer workload and lifestyle, farm facilities, milk contract and farmer mindset [42], and has the potential to influence farmer opinions on the attitudinal questions.

Multicollinearity diagnostics were run to complete the selection of farmer demographic and farming system variables to include as independent variables in the models. The final candidate models were compared using AIC to ensure the response to our original hypothesis is both informative and parsimonious, and a 5% significance level was the threshold for statistically significant contributions.

## Results and discussion

A total of 254 surveys were completed. There were 237 responses to the postal survey out of 909 posted to farmers (26% response rate; additional surveys were filled in online which for the target population and a 95% confidence level leads to a 5.22% margin of error in our results) and 11 surveys were returned stating that the dairy farm was no longer in business. A charitable donation of £508 was made to the Royal Scottish Agricultural Benevolent Institute. The data are available to view on the UK Data Service [43].

Descriptive statistics of demographic and farm system variables are shown in Table 1. The respondent median herd size was 160 cows. The average herd size in Scotland is currently 201 cows [11], which is similar to the mean of 206 in the sample. The sample average milk production per cow per year was very similar to the national average: 7966 litres compared to the UK average of 7825 litres [44] (when Scottish figures are not available, figures for the UK dairy industry will be used). Respondent calving pattern was similar to the UK dairy farmer population: 80% of respondents carried out year-round calving compared to 79% of the UK dairy population [42].

Very few farms, 2%, grazed cows all year around. The most common housing and grazing practice was 'summer grazing, winter housing with additional feed' operated by 41%, followed by 'summer grazing, winter housing with minimal additional feed', operated by 38% of farms. 51% of farms had expanded since 2015 and 33% planned to expand in the near future. This accords with the trend in Scotland of a continuing increase in farm size [45]. By far the most commonly reported route to past and future expansion was increasing cow numbers (84% for

**Table 1. Respondent descriptive statistics.**

| Responses (n) | Gender (%) | | Time in farming (%) | | | | |
|---|---|---|---|---|---|---|---|
| | **Male** | **Female** | **<10 years** | **10–20** | **20–30** | **>30** | |
| 254 | 96 | 4 | 4 | 8 | 21 | 67 | |
| **Cow numbers (n)** | | | | | | | |
| Median | Mean | Max | Min | IQR | | | |
| 160 | 206 | 1300 | 29 | 240–120 | | | |
| **Milk yield (litres)** | | | | | | | |
| Median | Mean | Max | Min | IQR | | | |
| 8000 | 7966 | 16000 | 3000 | 9000–68000 | | | |
| **Education (%)** | | | | | | | |
| GCSE equivalent | **A-level equivalent** | **Certificate** | **Diploma** | **Degree** | **Postgraduate degree** | | |
| 23 | 11 | 20 | 25 | 19 | 1 | | |
| **Ownership structure (%)** | | | | | | | |
| Owner | **Manager** | **Employee** | **Family** | **Partner** | **Other** | | |
| 92 | 2 | <1 | <1 | 5 | <1 | | |
| | | **Full time labour units** | | | | | |
| **Organic (%)** | **Conventional (%)** | Median | Max | Min | Interquartile range (IQR) | | |
| 6 | 94 | 3 | 17 | 0.5 | 4–2 | | |
| **Land owned** | | | | | | | |
| (n) | Median (ha) | Max (ha) | Min (ha) | IQR (ha) | | | |
| 222 | 141 | 809 | 9 | 200–98 | | | |
| **Land rented** | | | | | | | |
| (n) | Median (ha) | Max (ha) | Min (ha) | IQR (ha) | | | |
| 159 | 70 | 600 | 2 | 120–30 | | | |
| **Total land** | | | | | | | |
| (n) | Median (ha) | Max (ha) | Min (ha) | IQR (ha) | | | |
| 247 | 180 | 1293 | 18 | 280–120 | | | |
| **Calving pattern (%)** | | | | | | | |
| Year-round calving | **Spring calving** | **Autumn calving** | **Spring and autumn calving** | | | | |
| 80 | 5 | 5 | 10 | | | | |
| **Expanded since 2015 (%)** | **Plan to expand in future (%)** | | | | | | |
| 51 | 33 | | | | | | |
| | **Means of expansion since 2015 (%)** | | | | | | |
| **More land** | **More cows** | **More concentrate** | **Different breeds** | **Change calving** | **Change grass management** | **Improve health/ fertility** | **Partnership** |
| 23 | 84 | 20 | 3 | 9 | 24 | 40 | 0 |
| | **Means of future expansion (%)** | | | | | | |
| 14 | 77 | 17 | 5 | 5 | 24 | 54 | 4 |
| | **Housing and grazing system (%)** | | | | | | |
| **Year-round grazing** | **Summer grazing, winter housing with minimal additional feed** | **Summer grazing, winter housing with additional feed** | **Year-round housing some lactating cows** | **Year-round housing all lactating cows** | **Year-round housing all cows (including followers)** | | |
| 2 | 38 | 41 | 4 | 12 | 3 | | |

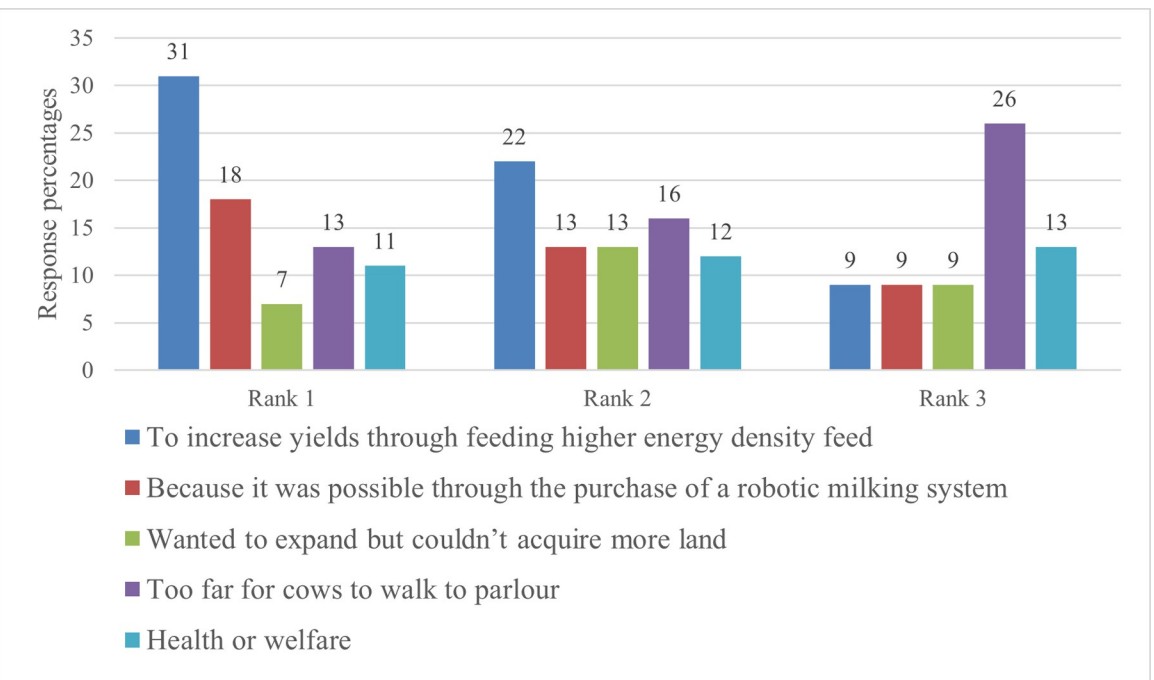

**Fig 1. Reasons for housing cows all year-round.**

past expansion, 77% for future expansion), followed by improving and health and/or fertility (40% for past expansion, 54% for future expansion).

Of the respondents, 19% housed some or all of the cows for all of the year. This figure is similar to previous studies for the UK, which showed 16% [12] and 23% [14] housed all or some of the cows all year round. Farmers who housed some or all of the cows all year-round were asked to rank the first, second and third reasons for doing so (Fig 1). 'Increasing production through feeding higher energy density feed' was the highest ranked reason, followed by 'because it was possible through the purchase of a robotic milking system', and 'too far for the cows to walk to the parlour'.

Logistics, including the distance cows have to walk from the parlour to the field, is cited in the literature as an important reason for why farmers move production indoors [15]. Research has also shown that being a high producing farmer is part of what it means to be a 'good farmer'–it bestows status in the farming community and is taken as a demonstration of skill [46]. Further research could explore the specific reasons for pursuing production increases through an indoor system: for instance, was increasing production seen as a matter of survival, a strategy to increase profits, a challenge, or a preference for a modern and progressive system?

The survey was conducted in 2018 when poor spring weather and a drought in summer put strain on dairy farmers and led them to feed more concentrate [47]. Thus, the number who fed additional feed may have been higher than if the survey was conducted in another year. The categories of grazing with minimal or additional concentrate feed may have been interpreted by farmers differently. This classification method was used in order to allow farmers to classify their own system and to make the survey easier to fill out rather than asking farmers to provide data of concentrate use per cow per year which they might not have had ready access to while filling in the survey. In addition, farmers change concentrate feeding year on year based on weather and milk prices, so asking for concentrate use in one year may not be useful measure [48].

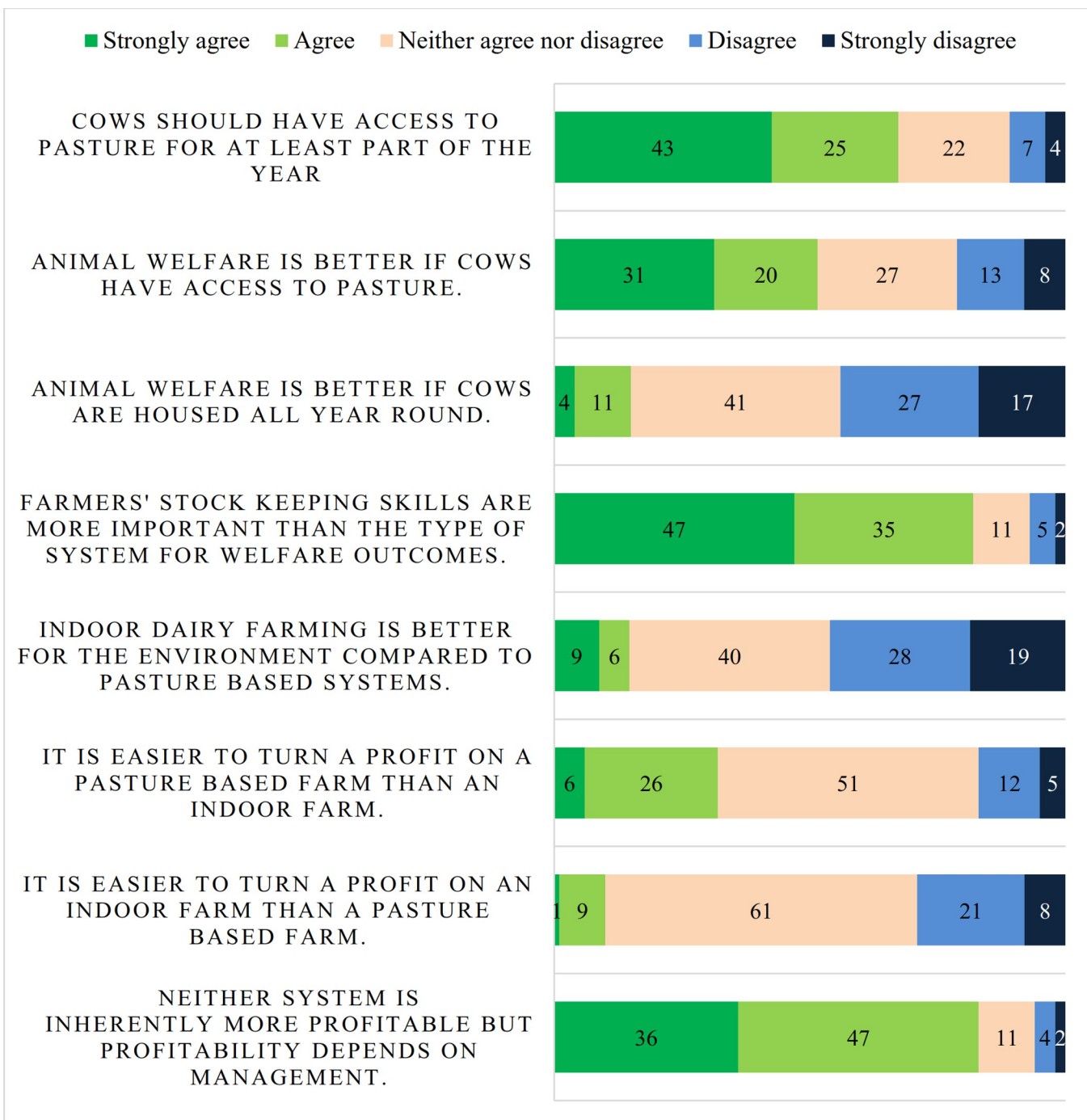

**Fig 2. Attitudes towards production systems.**

A summary of the responses to attitudinal questions on different production systems is shown in Fig 2. Respondents were asked to indicate how much they agreed with the statement 'Cows should have access to pasture for at least part of the year' and 68% agreed (strongly 43% and agreed 25%). Respondents were asked to respond to the statement 'Animal welfare is better if cows have access to pasture for part of the year' and 51% agreed (strongly 31% and agreed

20%). Few respondents saw animal welfare as better on indoor systems: 15% agreed (4% strongly agree, 11% agree). Most respondents endorsed the statement 'The farmer's stock keeping skills are more important than the type of system (indoor or pasture-based) for animal welfare': 82% (strongly agree 47% and agree 35%). Few respondents endorsed the statement 'Indoor dairy farms are better for the environment compared to a pasture-based system': 15% agreed (9% strongly agree, 6% agree).

The figure of 68% of respondents who strongly agreed or agreed that cows should have access to pasture was lower than surveys with the public [6, 7, 27]. The endorsement of access to pasture among respondents was nevertheless surprisingly high given the dominant discourse in the UK industry that it does not matter if cows graze or not, but economic, environmental and animal welfare outcomes depend on management rather than system [8]. This view that management is more important than system for welfare outcomes was strongly endorsed by respondents. However, this emphasis on the importance of management did co-exist with the majority view that cows should have access to pasture, and half of respondents agreed welfare was better in a grazing system. In the past, the question of public opposition to year-round housing of dairy cows has been framed as the industry accepting indoor dairy farming and the practice being rejected by the public because of a deficit in information [28]. The results from this study suggest that the picture is more complicated. Previous surveys on farmer's attitudes towards grazing in Germany and Denmark focused on the advantages and disadvantages of grazing and did not ask a normative question about whether cows *should* graze [29–31]. Studies which compared the views of farmers and citizens found that citizens cared more about cows grazing than did farmers [33–35]. The results of this study are therefore novel in showing that the majority of farmer respondents agreed cows should graze and over half agreed welfare was better if cows graze.

Agricultural organisations and policy makers should take into account that grazing is important to a large proportion of Scottish farmers. This could be done through measures to safeguard farmers' ability to graze through research and advice about grazing in the face of the climate crisis which will make grazing more challenging [3]. Ensuring that environmental assessments of dairy farmers are not calculated in such a way that they penalise farmers who graze [26]. And safeguarding farmers' ability to keep grazing in the face of ongoing restructuring where smaller farms may leave the industry and remaining farms tend to get bigger. This could be done through for instance supply chains which market grazed milk and pay farmers a premium for grazing. There are concerns within the dairy industry in the UK that the marketing of dairy produce based on access to pasture will divide the sector and lead to the further vilification of indoor farming among the public [8]. The results in this survey suggest that there are differences of opinion in the dairy sector in Scotland, indicating that valued based discussion addressing indoor and pasture-based systems could serve to ameliorate rather than exacerbate tensions [49]. Given that more respondents agreed cows should have access to pasture than agreed that animal welfare was better if cows had access to pasture, this suggests that the farmers' views go beyond animal welfare concerns. Other values could be at play including the importance of tradition, landscape, agricultural heritage, aesthetics and connection to the natural world.

The variables selected for the cluster analysis were those that were directly relevant to the research question and had a small number of missing values, including: time in farming, educational level, ownership, number of cows, total land operated, yield (l), number of labour units, calving pattern, grazing system, attitudinal questions about grazing; welfare in a grazing system; welfare in an indoor system; welfare outcomes being independent of system; and the environmental sustainability of an indoor system. Results from the cluster analysis showed two main clusters of farmers (cluster 1 with 164 farms, and cluster 2 with 89 farms) which

**Table 2. Pasture and system welfare responses according to cluster membership (%\*).**

| Education (%) | GCSE equivalent | A-level equivalent | Certificate | Diploma | Degree | Postgraduate degree |
|---|---|---|---|---|---|---|
| Cluster 1 | **2.0** | **6.5** | **26.0** | **35.7** | **27.9** | **1.9** |
| Cluster 2 | **68.6** | **22.9** | **7.1** | **1.4** | **0.0** | **0.0** |
| **Time in farming (%)** | **<10 years** | **10–20** | **20–30** | **>30** | | |
| Cluster 1 | **6.7** | **11.6** | **27.4** | **54.3** | | |
| Cluster 2 | **0.0** | **1.1** | **7.9** | **91.0** | | |
| **Pasture welfare (%)** | **Strongly agree** | **Agree** | **NA/ND** | **Disagree** | **Strongly disagree** | |
| Cluster 1 | 27.16 | 19.14 | 29.63 | 14.20 | 9.88 | |
| Cluster 2 | 38.64 | 21.59 | 22.73 | 11.36 | 5.68 | |
| **System welfare (%)** | **Strongly agree** | **Agree** | **NA/ND** | **Disagree** | **Strongly disagree** | |
| Cluster 1 | 53.09 | 30.86 | 9.88 | 4.32 | 1.85 | |
| Cluster 2 | 36.78 | 41.38 | 13.79 | 5.75 | 2.30 | |

\*Percentages might not add up to 100% due to rounding.

were significantly different in terms of their time in farming (p<0.001), their education levels (p<0.001) and their views on whether welfare was better if cows graze (p-value = 0.029) and whether welfare outcomes depend more on management than a grazing or indoor system (p-value = 0.021). Farmers in cluster 2 had spent more time in farming, they reached lower education levels, and they were moderately more likely to agree or strongly agree that welfare was better if cows had access to pasture, and slightly less more to disagree or strongly disagree that management was more important than system for determining welfare outcomes (see Table 2 for percentages in each of the response categories for those variables with statistically significant differences between the two clusters). Farmers who had spent longer in farming are more likely to be older farmers, who may be more in favour of a 'traditional' practice such as grazing. This is not consistent with findings from Germany where older farmers were less likely to have a positive attitude towards grazing [29].

A model was fitted to assess which factors have an effect on responses to the statement 'Cows should have access to pasture for at least part of the year' (Table 3). Respondents who operate a system involving grazing, as opposed to an indoor system, and respondents who had

**Table 3. 'Cows should have access to pasture for at least part of the year'–ordinal model summary results.**

| | Estimate (SE) | z value | Pr(>\|z\|) | 95% Confidence interval |
|---|---|---|---|---|
| Calving (Year-round contrast) | | | | |
| Spring calving | -0.523 (0.797) | -0.66 | 0.512 | -2.0850, 1.039 |
| Autumn calving | 1.180 (0.704) | 1.68 | 0.094 | -0.198, 2.560 |
| Spring & autumn calving | -0.208 (0.492) | - 0.42 | 0.673 | -1.174, 0.758 |
| Calving 'other' | 13.792 (928.829) | 0.01 | 0.988 | -1806.679, 1834.264 |
| Grazing (System involves grazing contrast) | | | | |
| Indoor system | 1.557 (- 0.451) | -3.45 | 0.001 | -2.441, -0.673 |
| Yield (l) | <0.000 (<0.000) | -3.30 | 0.000 | -0.001, 0.000 |
| Cows | -<0. 000 (<0.000) | -0.16 | 0.871 | -0.001, 0.002 |
| Time farming | -0.479 (0.166) | 2.89 | 0.004 | 0.154, 0.805 |
| Education | -0.026 (0.099) | -0.26 | 0.791 | -0.219, 0.167 |
| | | | | **Goodness of fit** |
| Deviance (df = 190) | | | | 517.752 |
| AIC | | | | 474.680 |

spent longer in farming were significantly more likely to agree, and respondents with higher yields were significantly more likely to disagree. This accords with finding with German [30], Danish [31] and Canadian [32] farmers, that people whose system involved grazing were more likely to have more positive attitudes towards grazing. The result of the model that farmers who have spent longer in farming are more likely to agree that cows should have access to pasture accords with the findings of the cluster analysis.

One third of the farmers (32%) agreed with the statement that it was easier to turn a profit on a pasture-based compared to an indoor farm (strongly 6% and agreed 26%) while 10% agreed (strongly 1% and agreed 9%) that it was easier to turn a profit on an indoor system. The majority of respondents endorsed the view that profitability was more about management than system type, 83% agreed (strongly agreed 36% and agreed 47%) with the statement 'Neither system is more profitable, but profitability depends on management'. As described in the introduction, there is a marginal view in the UK industry that grass-based systems are lower cost and easier to manage than higher feed input and indoor systems [8], which was echoed by the 32% of respondents who agreed or strongly agreed that it's easier to turn a profit on a pasture based farm. But respondents overwhelmingly agreed that management was more important than systems differences, echoing the mainstream view from the industry [8].

A model was fitted to assess which factors have an effect in the responses to the statement 'It's easier to turn a profit on a pasture-based farm than an indoor farm' (Table 4). Farmers who operated a system involving grazing were significantly more likely to strongly agree or agree with the statement. This accords with research that farmers are more likely to endorse the benefits of the system they operate [29–32].

Over half of respondents were either very satisfied (5%) or satisfied (47%) with how profitable their dairy farm is (Fig 3). Profitability is an issue in the Scottish dairy sector: a report by the Dairy Sector Climate Change Group states that in 2018–2019 only around 50% of Scottish dairy farms were profitable without subsidy, and with subsidy around 60% of farms were profitable, meaning the remaining 40% did not return a profit even with subsidy payment [26]. More than a third of respondents were either very satisfied (2%) or satisfied (33%) with their work life balance. More respondents were dissatisfied than satisfied (29% dissatisfied and 10% very dissatisfied) with their work life balance. This accords with findings that farmers in Scotland, and particularly livestock farmers, work longer hours than the average worker [50].

**Table 4. 'It's easier to turn a profit on a pasture-based farm than an indoor farm'–ordinal model summary results.**

|  | Estimate (SE) | z value | Pr(>\|z\|) | 95% Confidence interval |
|---|---|---|---|---|
| Calving (Year-round contrast) |  |  |  |  |
| Spring calving | 0.001 (0.751) | -0.25 | 0.804 | -1.659, 1.285 |
| Autumn calving | -0.087 (0.571) | -0.15 | 0.878 | -1.206, 1.031 |
| Spring & autumn calving | 0.968 (0.492) | 2.01 | 0.044 | 0.025, 1.911 |
| Calving 'other' | 1.662 (1.631) | -1.02 | 0.308 | -1.534, 4.858 |
| Grazing (System involves grazing contrast) |  |  |  |  |
| Indoor system | -1.194 (0.440) | -2.72 | 0.007 | -2.056, -0.332 |
| Yield (l) | -<0.000 (<0.000) | -0.15 | 0.882 | <0.000, <0.000 |
| Cows | -<0.000 (<0.000) | -1.00 | 0.315 | -0.001, 0.001 |
| Time farming | 0.139 (0.167) | 0.83 | 0.407 | -0.189, 0.467 |
| Education | 0.071 (0.095) | 0.75 | 0.456 | -0.115, 0.257 |
|  |  |  |  | **Goodness of fit** |
| Deviance (df = 192) |  |  |  | 572.105 |
| AIC |  |  |  | 528.906 |

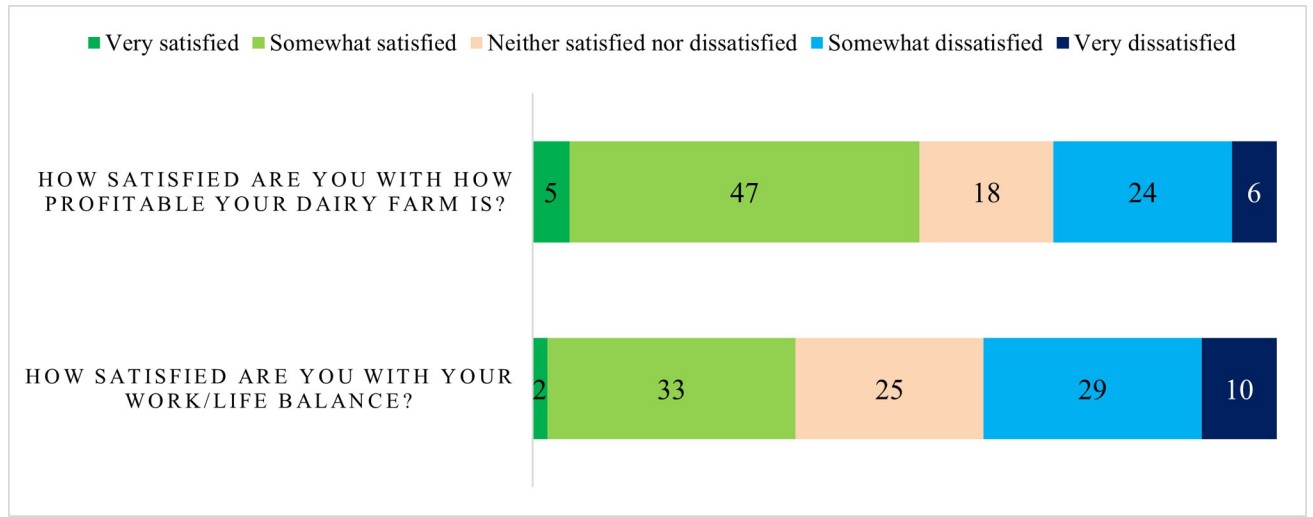

**Fig 3. Satisfaction with profitability and work life balance.**

Models were fitted to assess which factors have an effect on satisfaction with profits and work life balance (Table 5). None of the factors were found to have a statistically significant effect on satisfaction with profit at 5% significance level. Respondents who operated an autumn and spring calving system were significantly more likely to be satisfied with work life balance. The levy body the Agricultural and Horticulture Board Dairy have outlined the lifestyle benefits of block calving [42], which may be borne out by these results.

Overall, there is a significant moderate positive correlation (r = 0.5351 p<0.001) between satisfaction with profitability and work-life balance. This may be because more profitable farms also involved a better work life balance for the respondent. Or because it's a self-

**Table 5. Satisfaction with profit (left) and with work life (right)–ordinal model summary results.**

| | Satisfaction with profit | | | | Satisfaction with work life | | | |
|---|---|---|---|---|---|---|---|---|
| | Estimate (SE) | z value | Pr(>\|z\|) | 95% Confident interval | Estimate (SE) | z value | Pr(>\|z\|) | 95% Confidence interval |
| Grazing (System involves grazing contrast) | | | | | | | | |
| Indoor system | 0.096 (0.415) | 0.23 | 0.818 | 0.909, 0.718 | 0.579 (0.425) | 1.36 | 0.172 | -1.411, 0.259 |
| Calving (Year-round contrast) | | | | | | | | |
| Spring calving | 0.356 (0.697) | 0.51 | 0.610 | 1.722, -1.010 | -0.086 (0.695) | -0.12 | 0.902 | -1.447, 1.276 |
| Autumn calving | 0.036 (0.549) | 0.07 | 0.947 | 1.108, -1.035 | 0.116 (0.561) | 0.21 | 0.836 | --0.98, 1.215 |
| Spring & autumn calving | 0.889 (0.524) | 1.69 | 0.090 | 1.916, -0.139 | 1.568 (0.514) | 3.05 | 0.002 | −0.560, 2.576 |
| Calving 'other' | -1.432 (1.617) | -0.89 | 0.376 | 1.738, -4.602 | -15.138 (664.531) | -0.02 | 0.982 | -1317.597, 1287.321 |
| Yield (l) | 0.000 (0.000) | 1.32 | 0.186 | <0.000, <-0.000 | 0.000 (0.000) | 1.45 | 0.147 | <-0.001, <0.001 |
| Cows | 0.001 (0.001) | 0.56 | 0.576 | 0.003, -0.002 | 0.002 (0.001) | 1.70 | 0.089 | 0.0035, <-0.001 |
| Time farming | 0.014 (0.163) | 0.09 | 0.931 | 0.333, -0.305 | 0.152 (0.151) | -1.01 | 0.313 | -0.449, 0.144 |
| Education | 0.010 (0.094) | 1.07 | 0.287 | 0.284, -0.084 | -0.067 (0.092) | -0.73 | 0.466 | -0.247, 0.113 |
| | | | | Goodness of fit | | | | Goodness of fit |
| Deviance (df = 191 \| 191) | | | | 603.421 | | | | 604.251 |
| AIC | | | | 560.285 | | | | 561.116 |

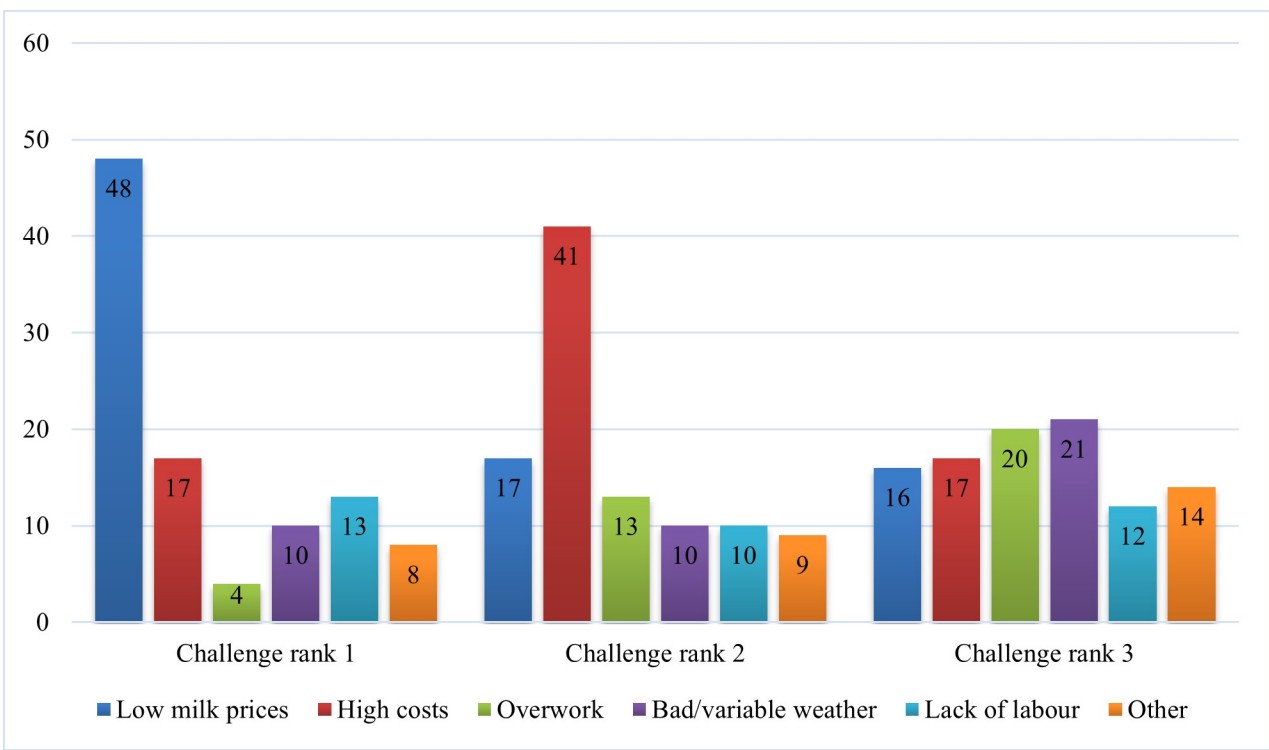

**Fig 4. Ranking of challenges facing dairy farmers.**

reporting measure it may be a reflection of the respondents' perspective: those who were more satisfied with profits were also more satisfied with work life balance.

Respondents were asked about the biggest challenge facing dairy farmers (Fig 4). The most commonly first ranked challenge was low milk prices (48%). The second most common challenge ranked first was high costs (17%). High costs were the most commonly ranked second biggest challenge (41%) followed by low milk prices (17%). Concerns about costs and prices echo media and industry discourse: the National Farmers Union of Scotland released a briefing stating that processors had an unequal amount of power in the dairy supply chain and risks and rewards were unfairly distributed [51]. The Scottish Government commissioned a report on dairy contracts in Scotland and recommended measures to increase farmer representation power and reduce the volatility of milk prices [52].

## Conclusion

This paper assessed the views and practices of Scottish dairy farmers relating to grazing and indoor systems. Over two thirds of respondents agreed cows should graze for part of the year and over half agreed welfare was better if cows grazed. These views co-existed with views that profitability and animal welfare were more dependent on stock keeping and management than the type of system. The endorsement of grazing is nevertheless surprisingly high given previous research showed that UK dairy industry stakeholders maintained that differentiating between dairy systems based on whether they grazed or not was not helpful.

The results showed that respondents whose system involved grazing and respondents who had spent longer in farming were more likely to believe that cows should have access to pasture, and less likely to agree that management was more important than the system for

determining animal welfare outcomes. This suggests that respondents were more likely to endorse a system that they operate and are more familiar with.

In terms of industry and policy recommendations, the research suggests that measures should be taken to safeguard farmers' ability to graze through for instance research and advisory support on grazing; ensuring different systems are not penalised in the development of dairy sector environmental measures and recommendations; and potentially supply chains that financially rewards farmers for grazing.

## Supporting information

**S1 Text. Survey posted to Scottish dairy farmers.**
(DOCX)

## Acknowledgments

The authors would like to thank the farmers who filled in the survey and the anonymous reviewers for their helpful comments.

## Author Contributions

**Conceptualization:** Orla K. Shortall.

**Data curation:** Orla K. Shortall, Altea Lorenzo-Arribas.

**Formal analysis:** Orla K. Shortall, Altea Lorenzo-Arribas.

**Funding acquisition:** Orla K. Shortall.

**Investigation:** Orla K. Shortall, Altea Lorenzo-Arribas.

**Methodology:** Orla K. Shortall, Altea Lorenzo-Arribas.

**Project administration:** Orla K. Shortall.

**Resources:** Orla K. Shortall, Altea Lorenzo-Arribas.

**Software:** Orla K. Shortall, Altea Lorenzo-Arribas.

**Validation:** Orla K. Shortall, Altea Lorenzo-Arribas.

**Visualization:** Orla K. Shortall, Altea Lorenzo-Arribas.

**Writing – original draft:** Orla K. Shortall, Altea Lorenzo-Arribas.

**Writing – review & editing:** Orla K. Shortall, Altea Lorenzo-Arribas.

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
