## [Decision Letter · Decision Letter 0]

27 Sep 2021

PONE-D-21-13820Dairy farmer practices and attitudes relating to pasture-based and indoor production systems in ScotlandPLOS ONE

Dear Dr. Shortall,

Thank you for submitting your manuscript to PLOS ONE. After careful consideration, we feel that it has merit but does not fully meet PLOS ONE’s publication criteria as it currently stands. Therefore, we invite you to submit a revised version of the manuscript that addresses the points raised during the review process.

I included some comments of my own review of the manuscript. If you can address the reviewers' suggestions plus mine, we will be able to revise your manuscript again.  Please submit your revised manuscript by Nov 11 2021 11:59PM. If you will need more time than this to complete your revisions, please reply to this message or contact the journal office at plosone@plos.org. Please include the following items when submitting your revised manuscript:A rebuttal letter that responds to each point raised by the academic editor and reviewer(s). You should upload this letter as a separate file labeled 'Response to Reviewers'.A marked-up copy of your manuscript that highlights changes made to the original version. You should upload this as a separate file labeled 'Revised Manuscript with Track Changes'.An unmarked version of your revised paper without tracked changes. You should upload this as a separate file labeled 'Manuscript'.

We look forward to receiving your revised manuscript.

Kind regards,

Luis Alonso Villalobos

Academic Editor

PLOS ONE

Additional Editor Comments (if provided):

Dear Dr Orla Shortall

We have received the comments from the reviewers and because the document needs to be improved is that I am suggesting a major revision. Besides the comments of the reviewers (shown below in this email), I am also giving some suggestions and edits from myself. Overall, the manuscript provides information that can be of potential use for the Scottish Dairy Sector, however, the importance of your findings needs to be addressed in the document and, especially, in the conclusions.

My comments are:

L45: period of housing cows in winter for temperate countries. An indoor...

L54: ....cows indoor for most of the year

L61: The current debate about the animal...

L62:...round, is explained by evidence suggesting that indoor dairy farming....

L86-91: very long sentence.

L140: ....why they had chosen this system.

L182: ....were asked the first, second and third reasons for moving production indoors all-year round (Fig. 1)

L185:...acquiring more land. (delete the rest of the sentence)

L191-193: wording

L194-201: this should be above the text about Fig. 1

L198: .....in farm size (40). The most commonly...

L199: ...and future expansion was by far increasing cow ...

L204-205: in what sense what this interpreted differently by farmers?

L213: ...and 68% agreed (strongly 43% and agreed 25%)

L215:...and 51% agreed (strongly 31% and agreed 20%)

L218:... 82% (strongly agree 47% and agree 35%)

L236: Scotland, indicating that valued based....

L242-250: very long sentence

L250: .....or indoor system (p-value=0.021). Farmers in cluster 2 had spent...

L260:.....at least part of the year (Table3). Respondents

L270-274: One third of the farmers (32%) agreed to the statement that it was easier to turn a profit on a pasture-based compared to an indoor farm (strongly 6% and agreed 26%) while 10% agreed (strongly 1% and agreed 9%) that it was easier to turn a profit on an indoor system. The majority of the respondents endorsed the view that profitability was more about management than system type (strongly agreed 36% and agreed 47%) with the statement 'Neither system is more profitable but profitability depends on management'.

L276: As it was described...

L276-277: data cited are not shown

L281: ...than an indoor farm' (Table 4). Farmers....

L288: At least half of respondents were either...

L289: ...farm is (Fig. 3). More than one third of respondents were either....

L292:...than the average worker (44). Delete the following sentence.

L295: ...balance (Table 5). None of the ...

L298: avoid the use of acronyms

L302:....This may be a reflection of the farm that more profitable....

L306: Respondents were asked to rank the biggest challenge facing dairy farmers (Fig 4). The most commonly...

L306-309: This part should be further discussed as both factors coincide as the main concerns of farmers.

Conclusions: Conclusions should not include references. Also, the conclusions should be more concise and be summarized so as not to repeat the same aspects that the discussion.

L316-321: both sentences were mentioned earlier

Journal Requirements:

2. We note you have included a table to which you do not refer in the text of your manuscript. Please ensure that you refer to Table 5 in your text; if accepted, production will need this reference to link the reader to the Table.

Reviewers' comments:

Reviewer's Responses to Questions

**Comments to the Author**

1. Is the manuscript technically sound, and do the data support the conclusions?

Reviewer #1: Yes

Reviewer #2: Yes

2. Has the statistical analysis been performed appropriately and rigorously? 

Reviewer #1: Yes

Reviewer #2: Yes

3. Have the authors made all data underlying the findings in their manuscript fully available?

Reviewer #1: Yes

Reviewer #2: No

4. Is the manuscript presented in an intelligible fashion and written in standard English?

Reviewer #1: Yes

Reviewer #2: Yes

5. Review Comments to the Author

Reviewer #1: This paper addresses a current topic with a keen approach. The issue that is questioning livestock in the most acute way, its carbon footprint, is only slightly addressed; however, I have no doubt that these results will become one of the guidelines for a very up-to-date discussion in many countries. Other comments are included in the attached file.

Reviewer #2: This manuscript is based on collecting responses from dairy farmers on their practices and attitudes relating to type of dairy production systems. The paper is well-written. Here are my comments and suggestions:

My concern with this study is the representatives of the data generated. Since the response rate was only 26%, how can this represent every kind of farmer?

The introduction is well written, and authors come directly to the point about the necessity to have farmers opinion about grazing vs indoor dairy production systems. I would also like to see what authors would like to achieve with the responses observed in this study. In other words, why is this study important?

Conclusion should state the implications of the findings of this study. Getting producer responses is great but what is the use of this? To further discussion? Policy changes? This will also show the impact these kind of studies have in changing

public perception.

L170-171: Why do we have line about charitable donation? Are we missing the context?

L181-182: When citing UK references, please specify if similar questions were asked in UK survey. Does 16 and 23 % year-round housing meant it included both “All cows” and “some cows” in the options?

L185: Please explain what Rank 1, 2, and 3 meant since all the ranks include similar explanations. It will be easier if explained in the manuscript.

L193: With the existing data available, can authors speculate why Scottish dairy farmers are pushing for production increases through an indoor system?

L197: Aren’t factors including more concentrate, different breeds are included in more cows? How are they different?

L194: is there a reason some values are presented in percent while others (means of expansion) are presented as just simple numbers since percent values are discussed in the manuscript (L199).

L220: The impact of dairy production systems on environment needs background information on greenhouse gas emissions, soil C sequestration, etc. I am not sure getting responses without sharing the context serves the purpose. What are the implications of these responses when some farmers may not have enough knowledge on how to estimate environmental footprint?

L242: How are these clusters different? Like which cluster had higher education level, number of cows etc. I didn’t see this data in the table. It will be good for readers to compare the observations in Table 2 with Cluster characteristics.

L263: Indoor system was also significant in this table. It goes well with yield. Please discuss this variable as well.

L271: Please rephrase these lines because it shows that 32% and 10% strongly agreed not 6 and 1%.

L280: How does calving time influence whether or not farmers agree on profits from pasture-based farm. Please explain the rationale of this question in the survey.

Conclusion:

L212-214: This is not in agreement with the study objectives. While the general impression shared here is from public, this study is focused more on producer responses. More appropriate way would be to share views from previous producer surveys.

6. PLOS authors have the option to publish the peer review history of their article (what does this mean?). If published, this will include your full peer review and any attached files.

Reviewer #1: No

Reviewer #2: No

---

## [Author Response · Author response to Decision Letter 0]

27 Oct 2021

Dear Dr Orla Shortall

We have received the comments from the reviewers and because the document needs to be improved is that I am suggesting a major revision. Besides the comments of the reviewers (shown below in this email), I am also giving some suggestions and edits from myself. Overall, the manuscript provides information that can be of potential use for the Scottish Dairy Sector, however, the importance of your findings needs to be addressed in the document and, especially, in the conclusions.

Thank you for this point. We highlighted the importance of the study further in the results/discussion and the conclusion. 

My comments are:

L45: period of housing cows in winter for temperate countries. An indoor...

Added. 

L54: ....cows indoor for most of the year

We haven’t changed ‘more’ to most’ as suggested, because ‘most’ suggests more than half of the year, which isn’t indicated in the paper referenced. 

L61: The current debate about the animal...

L62:...round, is explained by evidence suggesting that indoor dairy farming....

We haven’t changed this sentence because we don’t think that the debate is explained by these papers. Debate suggests two sides and we wanted to show the reader that some sections of the industry contest the scientific evidence. 

L86-91: very long sentence.

We shortened the sentence. There is still a long sentence there, but it is a list of findings with semi-colons separating them, so we think it is readable.

L140: ....why they had chosen this system.

Changed. 

L182: ....were asked the first, second and third reasons for moving production indoors all-year round (Fig. 1)

Changed. 

L185:...acquiring more land. (delete the rest of the sentence)

Changed.

L191-193: wording

Wording was changed. 

L194-201: this should be above the text about Fig. 1

Text was moved. 

L198: .....in farm size (40). The most commonly...

Changed. 

L199: ...and future expansion was by far increasing cow ...

We would like to keep the current wording as we feel it reads better. Putting ‘by far’ just in front of ‘increasing cow numbers’ might read as if the ‘by far’ is part of the ‘increasing cow numbers’ clause, and it might take the reader a moment to digest the intended meaning. 

L204-205: in what sense what this interpreted differently by farmers?

Farmers might have interpreted the amount of concentrate that is ‘minimal’ or ‘additional’ differently. 

L213: ...and 68% agreed (strongly 43% and agreed 25%)

Changed. 

L215:...and 51% agreed (strongly 31% and agreed 20%)

Changed. 

L218:... 82% (strongly agree 47% and agree 35%)

Changed. 

L236: Scotland, indicating that valued based....

Changed. 

L242-250: very long sentence

Sentence was broken into two. 

L250: .....or indoor system (p-value=0.021). Farmers in cluster 2 had spent...

Changed.

L260:.....at least part of the year (Table3). Respondents

Changed. 

L270-274: One third of the farmers (32%) agreed to the statement that it was easier to turn a profit on a pasture-based compared to an indoor farm (strongly 6% and agreed 26%) while 10% agreed (strongly 1% and agreed 9%) that it was easier to turn a profit on an indoor system. The majority of the respondents endorsed the view that profitability was more about management than system type (strongly agreed 36% and agreed 47%) with the statement 'Neither system is more profitable but profitability depends on management'.

Changed. 

L276: As it was described...

We’d prefer to keep this wording as it is. We think that ‘As described in the introduction’ is acceptable English. 

L276-277: data cited are not shown

This refers to data cited in the paragraph above. This was clarified. 

L281: ...than an indoor farm' (Table 4). Farmers....

Changed. 

L288: At least half of respondents were either...

Changed. 

L289: ...farm is (Fig. 3). More than one third of respondents were either....

Changed. 

L292:...than the average worker (44). Delete the following sentence.

Changed. 

L295: ...balance (Table 5). None of the ...

Changed. 

L298: avoid the use of acronyms

Changed.

L302:....This may be a reflection of the farm that more profitable....

Changed. 

L306: Respondents were asked to rank the biggest challenge facing dairy farmers (Fig 4). The most commonly...

Changed. 

L306-309: This part should be further discussed as both factors coincide as the main concerns of farmers.

Costs and prices were discussed further. 

Conclusions: Conclusions should not include references. Also, the conclusions should be more concise and be summarized so as not to repeat the same aspects that the discussion.

L316-321: both sentences were mentioned earlier

We shortened the conclusion and took out references. We also included policy and industry recommendations. There is a degree of repetition with the discussion, but we consider the function of the conclusion to communicate to the reader the key take home messages. 

Journal Requirements:

We checked PLOS ONE’s style requirements. 

2. We note you have included a table to which you do not refer in the text of your manuscript. Please ensure that you refer to Table 5 in your text; if accepted, production will need this reference to link the reader to the Table.

A reference was put in the text to Table 5. 

A caption for the supporting information file was included at the end of the manuscript and the in-text citation was updated. 

Reviewers' comments:

Comments to the Author

1. Is the manuscript technically sound, and do the data support the conclusions?

Reviewer #1: Yes

Reviewer #2: Yes

2. Has the statistical analysis been performed appropriately and rigorously? 

Reviewer #1: Yes

Reviewer #2: Yes

3. Have the authors made all data underlying the findings in their manuscript fully available?

Reviewer #1: Yes

Reviewer #2: No

The survey data was accepted by the UK Data Service so a reference was included for the data. 

4. Is the manuscript presented in an intelligible fashion and written in standard English?

Reviewer #1: Yes

Reviewer #2: Yes

Reviewer's Responses to Questions

5. Review Comments to the Author

Reviewer #1: This paper addresses a current topic with a keen approach. The issue that is questioning livestock in the most acute way, its carbon footprint, is only slightly addressed; however, I have no doubt that these results will become one of the guidelines for a very up-to-date discussion in many countries. Other comments are included in the attached file.

Comments to “Dairy farmer practices and attitudes relating to pasture-based, high- input and indoor production systems in the UK and Ireland”.

This paper addresses a current topic with a keen approach. The issue that is questioning livestock in the most acute way, its carbon footprint, is only slightly addressed; however, I have no doubt that these results will become one of the guidelines for a very up-to-date discussion in many countries.

Line 18: and many others following below; lack of consistency, in some lines “year round” is used and in others “year-round”.

Thank you, all instances were changed to “year-round”. 

Line 30: should be “Scotland,”

The space after the comma was removed. 

Lines 57 to 63: this contradiction should not be left unsolved (lines 57 -60) “Year-round housing allows farms to … have greater control over the health … the cows; and (line 62) “Evidence has suggested that indoor dairy farming can result in worse health outcomes for cows”.

We see your point that this could be seen as a contradiction. Though we think claims to greater control over health through more human interventions in the cows’ health don’t necessarily contradict claims that year-round housing can cause more health problems compared to grazing. But to simplify we’ve changed the wording to ‘oversight of the health and activities of the cows.’

Lines 64 and 65: “cows’ emotional wellbeing” is more accurate than “cows’ subjective wellbeing”.

Changed.

Lines 67 and 68: “many call for more up to date research and greater understanding of the implications of year-round housing for animal health and welfare” Leaving this assessment unchallenged apparently ignores findings in the last few years based on solid research demonstrating the benefits to animal welfare from the access to pasture (e.g. Arnott G., Ferris C. P. &O’Connell N. E. (2017). Review: welfare of dairy cows in continuously housed and pasture-based production systems. Animal (2017), 11, 2: 261–273 doi:10.1017/S1751731116001336. Burow E., Rousing T., Thomsen P. T., Otten N. D. & Sørensen J. T. (2013). Effect of grazing on the cow welfare of dairy herds evaluated by a multidimensional welfare index. Animal 7:5, pp 834–842 doi:10.1017/S1751731112002297 Mee J. F. & Boyle L. A. (2020). Assessing whether dairy cow welfare is “better” in pasture-based than in confinement-based management systems. New Zealand Veterinary Journal, 68,3: 168-177.)

That’s a fair point, ending the paragraph on that sentence could be misleading. We’ve changed it to a reference to a qualitative study with key industry stakeholders where they denied the existence of welfare differences between systems. The point isn’t to put the research and industry claims on equal footing, but to show the reader the parameters of the debate. Thank you for the Burow et al. and Mee and Boyle reference which we have included. 

Lines 74 to 77: greenhouse gas emissions from the agricultural and transport systems involved in making feedstuffs available to housed cattle should not be excluded from such estimates, which otherwise are biased. 

We don’t want to get too bogged down in arbitrating on the different arguments made. The point of including them is not to highlight to the reader which is true of false, but to show the parameters of the debate in academia and the industry more widely. The Gerber et al. study referenced did include the production and transportation of feed in the life cycle analysis. 

Line 156-157: I appraise that the answer of the authors given to Reviewer #2 on this issue is uncomplete, since there are well accepted methods used to remove outliers.

The continuous variables in our data sets (number of cows, amount of land, milk yield and number of labour units) were assessed for the presence of outliers graphically through the use of boxplots that highlighted the presence of anomously high numbers (not consistent with the overall population) that were obvious consequences of input errors. A small number of errors were removed: maximum 6 per variable. 

Line 289: “farm is. 35% of” should be “farm is; 35% of”

This was already changed in line with the editor’s suggestion. 

Line 328: “they are operate” should be “they operate”.

Changed. 

Line 362: should be “Gonzalez-Mejia”

Changed. 

Line 430: should be “in biophysical”; should be of spring.

Changed. 

Reviewer #2: This manuscript is based on collecting responses from dairy farmers on their practices and attitudes relating to type of dairy production systems. The paper is well-written. Here are my comments and suggestions:

My concern with this study is the representatives of the data generated. Since the response rate was only 26%, how can this represent every kind of farmer?

When designing our survey’s sampling strategy on the basis of the size of our target population (909 as provided by the Scottish government) we aimed to achieve a realistic margin of error between 5% and 6% at a 95% confidence level, and we are satisfied with the achieved 5.22%. These figures were included in the results section. The response rate of 26% is consistent with the typical response rates of 20% to 30% reported in the survey literature (Yammarino et al, 1991) and with the fact that response rates to mail-based surveys have been reported to be declining in recent decades, and survey response rates for farmers have been acknowledged to be low overall (Glas et al, 2019, Pennings et al, 2002). Declining response rates have been reported by UK government agencies too (e.g., by the Wales Government in their Survey of Agriculture and Horticulture: https://gov.wales/june-survey-agriculture-and-horticulture-quality-report-html). 

We are also confident the sample demographics reflect well those of the general Scottish farming population as shown by our descriptive statistics of the sample in the second paragraph of the results and discussion section. 

References:

Joost M. E. Pennings, Scott H. Irwin and Darrel L. Good. (2002) Surveying Farmers: A Case Study. Review of Agricultural Economics, Vol. 24, No. 1, pp. 266-277.

Yammarino, F.J., Skinner, S.J. and Childers, T.L. (1991). Understanding Mail Survey Response Behavior: a Meta Analysis. Public Opinion Quarterly 55, 613-639.

Glas, Z.E., Getson, J.M., Gao, Y., Singh, A.S., Eanes, F.R., Esman, L.A., Bulla, B.R., 

 and Prokopy, L.S. (2019). Effect of Monetary Incentives on Mail Survey Response Rates for Midwestern Farmers. Society & Natural Resources, Volume 32, pp. 229-237.

The introduction is well written, and authors come directly to the point about the necessity to have farmers opinion about grazing vs indoor dairy production systems. I would also like to see what authors would like to achieve with the responses observed in this study. In other words, why is this study important?

Conclusion should state the implications of the findings of this study. Getting producer responses is great but what is the use of this? To further discussion? Policy changes? This will also show the impact these kind of studies have in changing public perception.

We recognise that more discussion of the importance of the study was needed. To this end policy and industry recommendations were included in the discussion, conclusion and abstract. 

L170-171: Why do we have line about charitable donation? Are we missing the context?

The charitable donation was a gesture of goodwill for people giving up their time to complete the survey, and an incentive to do so. This information was included. 

L181-182: When citing UK references, please specify if similar questions were asked in UK survey. Does 16 and 23 % year-round housing meant it included both “All cows” and “some cows” in the options?

Thanks, that’s a good point, I indicated that the percentages from other studies are of all or some of the cows. 

L185: Please explain what Rank 1, 2, and 3 meant since all the ranks include similar explanations. It will be easier if explained in the manuscript.

The wording used in the question was included in the text to make the differences clearer. 

L193: With the existing data available, can authors speculate why Scottish dairy farmers are pushing for production increases through an indoor system?

I don’t think we can speculate with the existing data. Increasing production is multifaceted because it includes economic, cultural and logistical factors. This type of question is more suited to qualitative research. 

L197: Aren’t factors including more concentrate, different breeds are included in more cows? How are they different?

We think they are separate because farmers could feed more concentrate to increase yields from their existing cows. They could also change to a different breeding strategy to for instance include more high yielding Holstein genetics without increasing the herd size. 

L194: is there a reason some values are presented in percent while others (means of expansion) are presented as just simple numbers since percent values are discussed in the manuscript (L199).

I assume that refers to Table 1. That was a mistake, thanks for spotting it, % was included in the row on means of expansion. 

L220: The impact of dairy production systems on environment needs background information on greenhouse gas emissions, soil C sequestration, etc. I am not sure getting responses without sharing the context serves the purpose. What are the implications of these responses when some farmers may not have enough knowledge on how to estimate environmental footprint?

This question was to assess farmers’ beliefs rather than their knowledge. Looking back, it would’ve been better to include environmental questions about grazing systems and the same question that environmental impact was more dependent on management than system, as was done for economics and animal welfare. We didn’t do this in order to reduce the length of the survey, because the shorter it is the more likely people are to finish filling it out. I (the main author) come from a school of social sciences that maintains views about scientific questions (and indeed the production of science itself), aren’t just informed by empirical knowledge but by cultural and social factors, identity and networks, among other things. So in that respect we see the question about the environment as no different from questions about animal welfare and economics: they can be assessed through different empirical means, but we were interested in the farmers’ views rather than their knowledge. We think that the environmental impacts of dairy farming are currently widely debated and while farmers will have different levels of knowledge, this doesn’t preclude them from having opinions, as is the case for their views on economics and animal welfare. If they didn’t have a definite opinion they could answer ‘neither agree nor disagree’. 

L242: How are these clusters different? Like which cluster had higher education level, number of cows etc. I didn’t see this data in the table. It will be good for readers to compare the observations in Table 2 with Cluster characteristics.

We have added frequencies to the table for the responses to education level and time in farming, which were the other variables on which the clusters significantly differed. We have not reported summaries for those variables included in the cluster analysis for which there were not statistical differences amongst clusters and therefore did not add relevant information. We have tried to make this clearer in the text too.

L263: Indoor system was also significant in this table. It goes well with yield. Please discuss this variable as well.

The significance of indoor systems was noted in the statement “Respondents who operate a system involving grazing”. It was specified in the text “as opposed to an indoor system”. This variable was discussed in relation to the findings of the studies by Becker et al. and Kristensen et al. 

L271: Please rephrase these lines because it shows that 32% and 10% strongly agreed not 6 and 1%.

This was rephrased in line with the editor’s suggestion. 

L280: How does calving time influence whether or not farmers agree on profits from pasture-based farm. Please explain the rationale of this question in the survey.

I didn’t quite understand this point. Did you mean justifying the inclusion of the calving variable in the models? We consider calving pattern as an important element of the farm system. We explained why in the methods section under ‘data analysis’. (Qualitative interviews carried out after the survey confirmed the importance of calving pattern. People operating a spring calving pattern saw themselves as in a different ‘camp’ with different beliefs and networks compared to all year round calvers. We haven’t published from this work yet so can’t include that.) 

Conclusion:

L212-214: This is not in agreement with the study objectives. While the general impression shared here is from public, this study is focused more on producer responses. More appropriate way would be to share views from previous producer surveys.

We changed the conclusion to make it shorter and better highlight the importance of the study and these lines were removed.

---

## [Decision Letter · Decision Letter 1]

21 Dec 2021

Dairy farmer practices and attitudes relating to pasture-based and indoor production systems in Scotland

PONE-D-21-13820R1

Dear Dr. Shortall,

We’re pleased to inform you that your manuscript has been judged scientifically suitable for publication and will be formally accepted for publication once it meets all outstanding technical requirements.

Kind regards,

Luis Alonso Villalobos

Academic Editor

PLOS ONE

Additional Editor Comments (optional):

Dear author

We appreciate the timely manner in which you sent the revised version. The reviewers have recommended that your manuscript fulfills the criteria of PLOS ONE and it can be published. The reviewers acknowledged that you incorporated their suggestions and comments in the revised version of the manuscript.

Reviewers' comments:

Reviewer's Responses to Questions

**Comments to the Author**

1. If the authors have adequately addressed your comments raised in a previous round of review and you feel that this manuscript is now acceptable for publication, you may indicate that here to bypass the “Comments to the Author” section, enter your conflict of interest statement in the “Confidential to Editor” section, and submit your "Accept" recommendation.

Reviewer #1: All comments have been addressed

Reviewer #2: All comments have been addressed

2. Is the manuscript technically sound, and do the data support the conclusions?

Reviewer #1: Yes

Reviewer #2: Yes

3. Has the statistical analysis been performed appropriately and rigorously? 

Reviewer #1: Yes

Reviewer #2: Yes

4. Have the authors made all data underlying the findings in their manuscript fully available?

Reviewer #1: Yes

Reviewer #2: Yes

5. Is the manuscript presented in an intelligible fashion and written in standard English?

Reviewer #1: Yes

Reviewer #2: Yes

6. Review Comments to the Author

Reviewer #1: You have addressed properly my comments and those issued by Reviewer Number 2 and the Editor, hence I consider that you have made the manuscript acceptable for publication.

Reviewer #2: I appreciate authors addressing all of my comments. This paper has good information and will be very useful for readers.

7. PLOS authors have the option to publish the peer review history of their article (what does this mean?). If published, this will include your full peer review and any attached files.

Reviewer #1: No

Reviewer #2: No

---

## [Editor Report · Acceptance letter]

25 Jan 2022

PONE-D-21-13820R1 

Dairy farmer practices and attitudes relating to pasture-based and indoor production systems in Scotland 

Dear Dr. Shortall:

I'm pleased to inform you that your manuscript has been deemed suitable for publication in PLOS ONE. Congratulations! Your manuscript is now with our production department. 

Kind regards, 

on behalf of

Dr. Luis Alonso Villalobos 

Academic Editor

PLOS ONE